# Association between Psoriasis and Renal Functions: An Integration Study of Observational Study and Mendelian Randomization

**DOI:** 10.3390/biomedicines12010249

**Published:** 2024-01-22

**Authors:** Yuxuan Tan, Zhizhuo Huang, Haiying Li, Huojie Yao, Yingyin Fu, Xiaomei Wu, Chuhang Lin, Zhengtian Lai, Guang Yang, Chunxia Jing

**Affiliations:** 1Department of Epidemiology, School of Medicine, Jinan University, No. 601 Huangpu Ave. West, Guangzhou 510632, China; 2Department of Pathogen Biology, School of Medicine, Jinan University, No. 601 Huangpu Ave. West, Guangzhou 510632, China

**Keywords:** psoriasis, renal function, cross-sectional study, GWAS summary statistics, mendelian randomization (MR)

## Abstract

Psoriasis is an autoimmune-mediated disease with several comorbidities in addition to typical skin lesions. Increasing evidence shows the relationships between psoriasis and renal functions, but the relationship and causality remain unclear. We aimed to investigate the associations and causality between psoriasis and four renal functions, including the estimated glomerular filtration rate (eGFR), blood urea nitrogen (BUN), urine albumin to creatinine ratio (UACR), and chronic kidney disease (CKD). For the population-based study, we analyzed the National Health and Nutrition Examination Survey (NHANES) data from five cycles (2003–2006 and 2009–2014) on psoriasis and renal functions. Subgroup analyses were conducted among different categories of participants. Meanwhile, a bidirectional two-sample Mendelian randomization (TSMR) study in European populations was also performed using summary-level genetic datasets. Causal effects were derived by conducting an inverse-variance weighted (MR-IVW) method. A series of pleiotropy-robust MR methods was employed to validate the robustness. Multivariable MR (MVMR) was conducted to complement the result when five competing risk factors were considered. A total of 20,244 participants were enrolled in the cross-sectional study, where 2.6% of them had psoriasis. In the fully adjusted model, participants with psoriasis had significantly lower eGFR (*p* = 0.025) compared with the healthy group. Individuals who are nonoverweight are more likely to be affected by psoriasis, leading to an elevation of BUN (*P*_int_ = 0.018). In the same line, TSMR showed a negative association between psoriasis and eGFR (*p* = 0.016), and sensitive analysis also consolidated the finding. No causality was identified between psoriasis and other renal functions, as well as the inverse causality (*p* > 0.05). The MVMR method further provided quite consistent results when adjusting five confounders (*p* = 0.042). We detected a significant negative effect of psoriasis on eGFR, with marginal association between BUN, UACR, and CKD. The adverse of psoriasis on the renal should merit further attention in clinical cares.

## 1. Introduction

Psoriasis is a chronic, inflammatory, and autoimmune-mediated skin disease affecting 2–4% of the population of Western countries [1]. According to the Global Burden of Disease (GBD) report, there were approximately 4.6 million incident cases of psoriasis worldwide in 2019 [2], causing a heavy public health burden.

The systemic inflammatory response caused by psoriasis makes the comorbidities of patients very diverse [3,4], which may involve multiple organs [5,6]. The kidney is particularly susceptible to damage caused by autoimmune conditions [7]. Recently, the concept of ‘psoriatic nephropathy’ has been proposed based on case reports [8]. The prevalence of proteinuria in patients with psoriasis has been reported to be 22–42% [9]. The prevalence of chronic kidney disease (CKD) in patients with psoriasis ranges from 1% to 8% in different populations [10]. Animal experiments have also shown that psoriasis-like inflammation could damage renal function via the *TLR*/*NF-κB* signal pathway [11,12]. However, in an observational study of nearly 1 million participants, the prevalence of CKD in psoriasis patients (3.54%) was close to that of healthy controls (3.97%) [10], indicating the relationship between psoriasis and renal function still needs to be further established.

Previous epidemiological studies have reported kidney disease among patients with psoriasis, but the results are inconsistent [13]. Nested research from the United Kingdom suggested moderate to severe psoriasis is associated with an increased risk of CKD [14]. Another study that aimed to evaluate the risk of incident CKD and end-stage renal disease (ESRD) in Chinese people (*n* = 926,987) with psoriasis revealed a marginally increased risk for CKD after adjustment for confounders (HR 1.84; 95% CI 1.01–3.84) [15]. A recent investigation including nearly 140,000 participants from Israel showed no association between psoriasis and CKD in adults or children [16]. However, a cross-sectional study with 16,750 participants from Yin et al. shows no significant association between psoriasis and CKD, whereas a marginal causal relationship applying genetic methods [17]. Traditional study designs are easily affected by uncontrolled confounders, so better approaches and comprehensive studies should be conducted.

To compensate for the uncertainty of observational studies in establishing a causal relationship between psoriasis and renal function, Mendelian randomization (MR) is used as a powerful tool. The underlying assumption of MR is that humans are naturally assigned a genetic variant associated with a risk factor or susceptibility to a disease or do not inherit such a variant [18]. These genetic variants are more reliable because they do not change with other factors [19]. The MR approach is less susceptible to confounding and reverses causality bias with valid instrumental variables (IVs). Thus, we first evaluated the associations between psoriasis and four renal functions on cross-sectional data from the National Health and Nutrition Examination Survey (NHANES). Then, we employed a MR framework based on data published by the genome-wide association study (GWAS) to further validate the causal relationship between each pair of traits.

## 2. Materials and Methods

### 2.1. Study Design

The overall study design was compounded with an epidemiological observational study and Mendelian randomization analysis. We first explored the associations between psoriasis and renal function through an epidemiological analysis from the NHANES. Subgroup analyses were used to investigate the potential interaction. Then, bidirectional two-sample MR (TSMR) and multivariable MR (MVMR) analyses were conducted to investigate the causality based on results from GWAS. We used multiple MR methods with different model assumptions to assess the robustness of the results. Additionally, sensitivity analyses were further conducted in both epidemiological and MR analyses. All analyses were conducted under the Strengthening the Reporting of Observational Studies in Epidemiology (STROBE) guidelines.

### 2.2. Epidemiological Observational Study

#### 2.2.1. Data Source

In the present study, we utilized the NHANES data from 5 cycles (2003–2006 and 2009–2014) to investigate the relationships between psoriasis and renal functions. NHANES is a cross-national survey program that aims to evaluate the health, nutritional, and dietary status of the general population in the U.S. Details can be found in previous descriptions [20]. After merging the demographic variables, questionnaire information, and laboratory test data, 27,567 participants with ages over 20 in 5 selected NHANES cycles were obtained. We further excluded 3503 and 3138 participants with missing psoriasis data or not eligible for the exclusion criteria, respectively. Finally, a total of 20,944 participants were included in this study. Figure 1 illustrates the flowchart of participant selection.

#### 2.2.2. Measurements and Definitions

Self-reported psoriasis was defined through questionnaires. Participants who answered “Yes” to “have ever been told by a doctor or other health care professional that they had psoriasis” were considered a psoriasis patient. Quality of the interview and data input process was promised by a computer-assisted personal interviewing system [20]. Three continuous renal functions were derived from NHANES laboratory data as dependent variables: estimated glomerular filtration rate (eGFR) based on creatinine levels, urine albumin to creatinine ratio (UACR), and blood urea nitrogen (BUN). Of note, eGFR was calculated by the Chronic Kidney Disease Epidemiology Collaboration (CKD-EPI) equation [21]. We further defined CKD by an eGFR ≤ 60 mL/min per 1.73 m^2^ or UACR ≥ 30.0 mg/g, as a dichotomous outcome [22,23].

#### 2.2.3. Covariates

The directed acyclic graph (DAG) was used to identify the covariates between psoriasis and renal functions based on prior knowledge of confounders (Appendix A). We obtained sociodemographic (e.g., gender, age, race/ethnicity, poverty status (PIR), education, body mass index (BMI)), lifestyle (e.g., smoking status (serum cotinine), alcohol drinking status (per day)), and medical history (e.g., diabetes, hypertension) as covariates. Education levels were categorized into five levels as used by NHANES. Participants were categorized as white, black, Mexican American, and other races. PIR was divided into two groups (<1.0 and ≥1.0). Marital status was defined as following three groups: married and living with a partner, other status (including widowed, divorced, or separated), and never married. For lifestyles, serum cotinine was used to quantify smoking status, and questionnaires were used to determine the average number of alcohols consumed per day. Diabetes was defined as self-reported diabetes, using antidiabetic medications, or fitting the criteria from the American Diabetes Association (ADA) [24]. Hypertension was determined by average blood pressure measured three times.

### 2.3. Mendelian Randomization Analysis

#### 2.3.1. Summary-Level GWAS Data

The summary-level GWAS statistics of psoriasis were from the FinnGen consortium. FinnGen study is an ongoing project aiming to produce nearly complete genome variant data of 500,000 biobank participants using GWAS genotyping [25]. The seventh round FinnGen study included 6995 cases of psoriasis and up to 299,128 controls diagnosed by ICD-10 (L40). Summary statistics for renal function, including log-transformed eGFR, UACR, and BUN were obtained from the CKDGen Consortium (https://ckdgen.imbi.uni-freiburg.de, accessed on 28 May 2023). Overall, 567,460 and 243,029 participants based on 42 and 24 studies were included in the summary data from eGFR and Bun summary data [26]. UACR summary data were obtained from another meta-GWAS which contained 547,361 individuals in up to 18 studies [27]. Additionally, 41,395 participants were diagnosed with CKD, and 439,303 controls were used to further investigate the relationship between psoriasis and renal function [26]. The CKD was defined as a binary outcome for eGFR > 60 mL/min per 1.73 m^2^, according to the previous study [23]. Appendix A shows the characteristics of the GWAS data sources, all data were retrieved in 27 March 2023. The genetic background was limited to European ancestry factors.

#### 2.3.2. Selection of Instrument Variables

We select single-nucleotide polymorphisms (SNPs) associated with exposures for each analysis at the genome-wide significance threshold (*p* < 5 × 10^−8^) and strength threshold (*F*-statistic > 10). Briefly, PLINK clumping algorithm with a 5000 kb LD window was performed to ensure independence (r^2^ < 0.01). While matching the outcome summary GWAS data, the high-LD proxy SNP (r^2^ > 0.8) was used to replace the unmatched SNP. Each pair of the exposure and outcome datasets were harmonized, and palindromic SNPs with a minor allele frequency greater than 0.42 were further removed.

### 2.4. Statistical Analyses

#### 2.4.1. Cross-Sectional Study

According to the NHANES analytical guidelines, complex sampling design and sampling weights were considered in all statistical analyses. The medians and interquartile range (IQR) were presented for continuous variables and percentages for categorical variables. We also conducted the Chi-square, one-way ANOVA, and Kruskal–Wallis test for categorical, normal distribution, and skewed distribution variables, respectively, to compare basic information in participants with or without psoriasis. Notably, we performed log-transform for UACR and BUN because of the skewed distribution. The generalized linear model was used to assess the association between psoriasis and renal functions. Two models were constructed for the exploration: Model 1 was the crude model that only adjusted for age and gender. We further built Model 2 with education levels, family PIR, marital status, smoking status, alcohol use, BMI diabetes, and hypertension on the base of Model 1. Moreover, we calculated the interaction between psoriasis and confounders using the Wald statistic to investigate the influence of psoriasis on renal functions, as a subgroup analysis.

To add robustness to our findings, several sensitivity analyses were performed. First, in order to explore the association in a complete dataset, multivariate imputations (MI) were used for the missing covariates [28]. Second, considering the overall body inflammation differences caused by psoriasis, we additionally adjusted the systemic immune-inflammatory index (SII) in a fully adjusted model [29]. Third, we further excluded participants with CKD, diabetes, hypertension, and hyperlipidemia for analysis among participants without comorbidity (only in continuous renal functions). Finally, we excluded 2 cycles (NHANES 2011–2014) to investigate the robustness between survey cycles.

#### 2.4.2. TSMR and MVMR Analyses

Given the fact that inadequate adjustment for confounding factors can bias the association between psoriasis and renal functions, we further used bidirectional TSMR to assess the evidence of their causal relationship. Linkage disequilibrium score (LDSC) regression was performed to assess the SNP-based heritability (h^2^), revealing a genetic association between psoriasis and renal function. MR using the inverse-variance weighted (MR-IVW) method was conducted as the principal analysis, which uses a weighted linear regression of the SNP coefficients between exposure and outcome to estimate the effect of exposure on outcome [30]. Meanwhile, multi-method MR approaches, which included simple mode, weighted median, MR-Egger regression, MR-PRESSO, and MR-RAPS methods, were performed to assess the robustness of our findings [31]. Associations for renal function on psoriasis were also explored for reverse causality investigation.

In sensitivity analyses for MR, we tested for heterogeneity whether statistically significant (*p*  <  0.05) using Cochran’s Q statistic by MR-IVW. Funnel plots were used to depict the heterogeneity [4]. Then, the MR-Egger intercept test and Causal Analysis Using Summary Effect estimates (CAUSE) method were used to illustrate the potentially horizontal pleiotropy and correlated pleiotropy, respectively [19]. In addition, we conducted leave-one-out (LOO) cross-validation analyses and MR-PRESSO analysis to reveal the causal inference as well as outliers. Appendix A summarizes the schematic diagram of the MR framework.

To assess the direct effect of psoriasis on renal functions, we further conducted MVMR analyses. GWAS data for five potential confounders that might affect the causal relationships were collected: smoking, drinking, physical activities (PA), obesity, and type 2 diabetes (T2DM), according to previous studies [32,33,34]. Appendix A presents the demographic profiles of GWAS data related to these factors. For MVMR, we applied an extension of the MR-IVW method (MVMR-IVW) as the main results, and the MVMR-Egger and MVMR-CML were used as sensitivity analyses [35].

The statistical analyses were conducted within R version 4.2.1 (R Foundation for Statistical Computing, Vienna, Austria) and Python programming language version 3.7.1 (Python Software Foundation, Cambridge, MA, USA). A two-sided *p*-value < 0.05 was considered statistically significant.

## 3. Result

### 3.1. Population-Based Study

Table 1 presents a summary of participant characteristics by psoriasis status. Of these participants, 549 (2.6%) reported suffering from psoriasis and with a similar prevalence in males and females. Compared with the healthy controls, participants with psoriasis had a higher likelihood of being White, with higher BMI, and had a higher prevalence of diabetes and hypertension. In addition, the levels of eGFR were significantly lower among participants with psoriasis than in healthy controls, and BUN had a significantly higher level in psoriasis participants. The distribution of three continuous renal functions has been demonstrated in Appendix A.

Estimates and the 95% confidence interval (CI) were used to assess the relationship between psoriasis and three continuous renal functions, whereas odds ratio (OR) and 95% CI were used for binomial outcomes. Figure 2 depicts the results of GLM. In the fully adjusted model, there were negative relationships between psoriasis and eGFR (estimate (95% CI): −2.96 (−5.55, −0.38), *p* = 0.025). No significant associations were found in psoriasis on three other renal functions (*p* > 0.05). The Wald test showed an interaction between psoriasis and BMI in the relationship of psoriasis on BUN (*P*_int_ = 0.018), indicating the effect of psoriasis on BUN may be influenced by BMI (Appendix A).

The sensitivity analysis results of the population-based study are demonstrated in Appendix A. There is still a negative relationship between psoriasis and eGFR in the dataset after multiple imputation (estimate (95% CI): −2.44 (−4.58, −0.29), *p* = 0.030). The results remained robust after further adjustment for SII and the exclusion of several comorbidities. Moreover, the survey cycle exclusion analysis showed a consistent result.

### 3.2. MR Framework Analyses

We evaluated the heritability and genetic correlations based on SNP using the LDSC regression. The SNP-based heritability was low for psoriasis and four renal functions (Appendix A). There was no evidence of genetic correlations between psoriasis and renal functions (rg = −0.026 to 0.058, *p* > 0.05), indicating little or no impact from sample overlap in the datasets. We further extracted 31 SNPs for psoriasis and 22 to 253 SNPs for renal functions as IVs in this study. Appendix A show the information on the selected IVs in the forward and reverse directions, respectively. There was no evidence of weak instrument bias as the instrument strength was strong (*F*-statistic in forward and reverse MR analyses ranging from 27.6 to 676.4).

Figure 3 presents the main results of the bidirectional TSMR analyses. Strong evidence indicates a significant negative effect of psoriasis on eGFR (estimate (SE): −0.0017 (0.0007), *p* = 0.016) using the MR-IVW method. Other MR approaches consolidated the negative trends for psoriasis on eGFR. We found no evidence that psoriasis is associated with a UACR (Estimate (SE): −0.0049 (0.0032), *p* = 0.331), BUN (Estimate (SE): 0.0011 (0.0011), *p* = 0.331), and CKD (Estimate (SE): 0.0135 (0.0115), *p* = 0.233), which is in line with epidemiological observational analysis. Meanwhile, the reverse TSMR analyses did not show evidence that renal functions may affect psoriasis (Appendix A). We also illustrated the point estimates for each SNP, detailed results were presented in Appendix A. 

We performed extensive sensitivity analyses to validate the causal association between psoriasis on renal function. There was notable heterogeneity in eGFR (Q = 79.2; *p* < 0.001) and UACR (Q = 53.6; *p* = 0.007) across instrument SNP effects (Appendix A). The diagnosed funnel plots are depicted in Appendix A. The intercept estimated by MR-Egger did not reveal significant horizontal pleiotropy (Appendix A). Correlated pleiotropy is generally overlooked, we use the CAUSE method to consider whether there is a correlated pleiotropy. Appendix A shows each model’s scatter plot and each variant’s contribution to the Δ Expected Log Pointwise Posterior Density (ΔELPD) test statistic. There was limited evidence that the causal model fit the data better than the sharing model in eGFR and BUN, indicating that the correlated pleiotropy could not be discounted (Appendix A). MR-PRESSO identified six and five outliers in eGFR and UACR, respectively. After removing the outliers, the associations of psoriasis on renal functions stay (Table 2). In addition, LOO analysis also showed that no SNP point estimates influence the causal effect estimation (Appendix A).

The effects of SNPs applied in the MVMR method are summarized in Appendix A. After controlling for the effect of five confounding factors, significant evidence was shown for the direct effect of psoriasis on the risk of eGFR (MVMR-IVW, Estimate (SE): −0.002 (0.001), *p* = 0.042), as well as using the MVMR-CML approach (Estimate (SE): −0.002 (0.001), *p* = 0.005). However, a marginal causality was discovered between psoriasis and CKD (Estimate (SE): 0.037 (0.019), *p* = 0.048), while sensitivity analysis failed to consolidate the result (Figure 4). For other traits, the MVMR also showed no evidence of causal associations (*p* > 0.05).

## 4. Discussion

This study investigates the association between psoriasis and four renal functions using large-scale observational study data and a MR framework. In the cross-section study, psoriasis was associated with eGFR (estimate (95% CI): −2.44 (−4.58, −0.29), *p* = 0.030), while no evidence was showed in UACR, BUN, and CKD. Subgroup analyses revealed a positive trend between psoriasis and BUN in the nonoverweight group, indicating the effect may vary with BMI. Meanwhile, the results were further verified by TSMR analysis in the European population, demonstrating a negative causality of psoriasis on eGFR, where sensitivity analyses consolidated the result. No causal evidence of the effect of renal function on psoriasis in the reverse direction was found. In addition, MVMR illustrated a robust link for psoriasis on eGFR, when we included five confounders in the MR framework. The results collectively indicate that psoriasis might confer a declining eGFR, more attention should be paid to renal functions in psoriasis clinical practice.

Notwithstanding the relationship between psoriasis and renal function has been explored extensively over the past decade, the conclusions from observational studies are still inconclusive. A systematic review by Yang et al. indicated only patients with severe psoriasis were associated with chronic kidney disease (CKD) or end-stage kidney disease (ESRD) [13]. Another nationwide study with over 6 million participants indicated a lower level of eGFR in participants with psoriasis [9]. However, Conti et al. found that there was no significant difference in eGFR between psoriasis patients and healthy control [36]. Inconsistent results may be due to reverse causation, measurement error, or underlying bias, all intrinsic limitations of observational studies [37]. Notably, a very recent study by Yin et al. suggested psoriasis was not cross-sectionally associated with CKD, and a marginal effect for psoriasis on CKD was observed when applying the MR framework [17]. Nevertheless, specific renal function, subgroup, and MVMR analysis were lacking. Our study corroborated results from previous observational studies showing associations between psoriasis and renal functions, as well as adding new evidence with robustness.

Epidemiological observational study and bidirectional TSMR demonstrated a relationship between psoriasis and eGFR. Systemic inflammatory response mediated by psoriasis might be a reason. The pathogenesis of psoriasis is linked to activated T-cells and cytokines such as tumor necrosis factor-α (TNF-α) [3]. As a pro-inflammatory cytokine, TNF-α becomes elevated in chronic inflammatory states, which further alters renal hemodynamics and nephron transport [38]. Animal experiments also showed that high doses of TNF-α infusion (≥0.6 mg/kg) in healthy rats can induce a lower glomerular filtration rate and renal blood flow [39]. In addition, TNF-α can cause hypertension via stimulating the production of endothelin 1 and angiotensinogen [40]. Previous studies had shown uncontrolled high blood pressure may result in blood vessel damage, leading to narrowing of the arteries around the kidneys, which in turn causes a decrease in kidney filter ability [41,42]. Therefore, inflammation might play a central role in the relationship between psoriasis and renal functions.

Another plausible explanation for the decrease in eGFR among patients with psoriasis is the alteration of serum levels of various cytokines, particularly IL-17 produced by Th17 cells. Psoriasis is primarily a dendritic and T-cell-mediated disease with complex feedback loops from antigen-presenting cells [1]. Numerous studies have demonstrated elevated levels of IL-17 in psoriasis skin lesions patients [43,44], and the serum level of IL-17 is positively associated with the severity of psoriasis [45]. Moreover, the IL17 pathway has been linked to immune-mediated glomerulonephritis [46,47]. The immunohistochemical co-staining for the pan T-cell marker, CD3, and IL-17, showed that the tubulointerstitial infiltrates in mice with immune-mediated nephritis contained CD3^+^IL-17^+^ cells, suggesting an essential role for IL-17 in immune-mediated nephritis [48]. Patients with nephritis generally have a lower glomerular filtration rate [49,50]. Therefore, the high levels of serum IL-17 in psoriatic patients may induce kidney inflammation and decrease eGFR. Further studies should focus on the possible underlying mechanism and role of pro-inflammatory cytokines between psoriasis and renal dysfunction.

This study possesses several strengths. We comprehensively investigate the relationship between psoriasis and four renal functions using both population-based data and MR framework. Although the cross-sectional study designs for NHANES limited causal inferences, the applied complementary MR approaches yielded homogeneous results and minimized potential confounding and reverse causation. Extensive sensitivity analyses highlighted the causality.

However, we must admit there are some limitations. First, even though the subgroup analyses were performed in the observational study, the TSMR analyses were based on summary-level data, so stratified results could not be assessed. Second, the dataset used in observational studies is relatively outdated, which might not be able to provide a more comprehensive understanding of the disease trends and potential confounders. Third, GWAS summary data is restricted to the European population, which reduces the generalizability of our findings. Fourth, abnormal renal function is associated with the severity of psoriasis, for which there are no data on the severity of psoriasis. We did not investigate different types of psoriasis due to low statistical power. In addition, conventional therapies, such as cyclosporine (Cya) and methotrexate (MTX), often result in transient impairment of renal function during psoriasis treatment [51]. Individual-level GWAS data and cohort studies were needed to investigate these associations in the future.

## 5. Conclusions

In summary, both the epidemiological study and MR framework suggested psoriasis is causally associated with a lower level of eGFR, whereas renal functions have no reverse causality on psoriasis. The early detection of renal dysfunction in psoriasis patients is an extremely important measure for outcome prevention. Concerning the causal relationship between psoriasis and renal function decrease, multidisciplinary shared care plans and complementary therapeutic solutions are necessary for better control of this interdependent pathogenic topic.

## Figures and Tables

**Figure 1 biomedicines-12-00249-f001:**
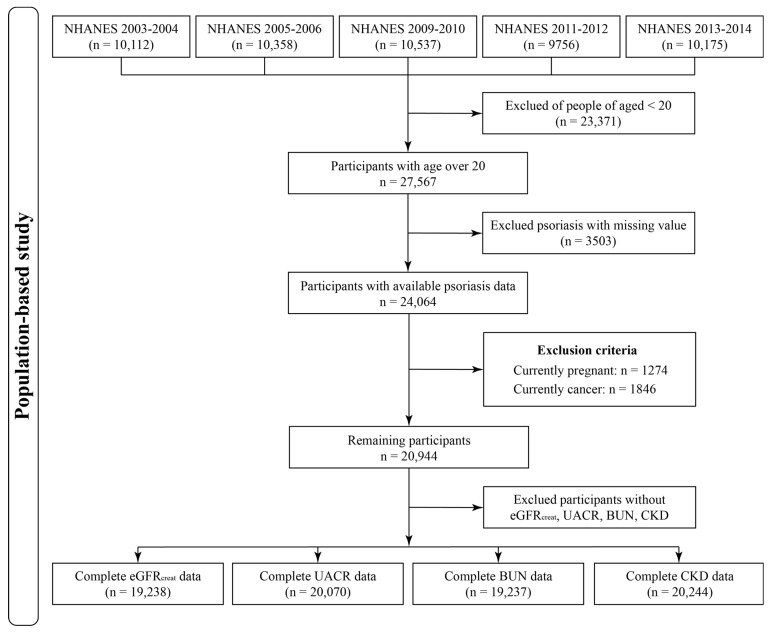
Flowchart of participants selection from the 5 NHANES cycles.

**Figure 2 biomedicines-12-00249-f002:**
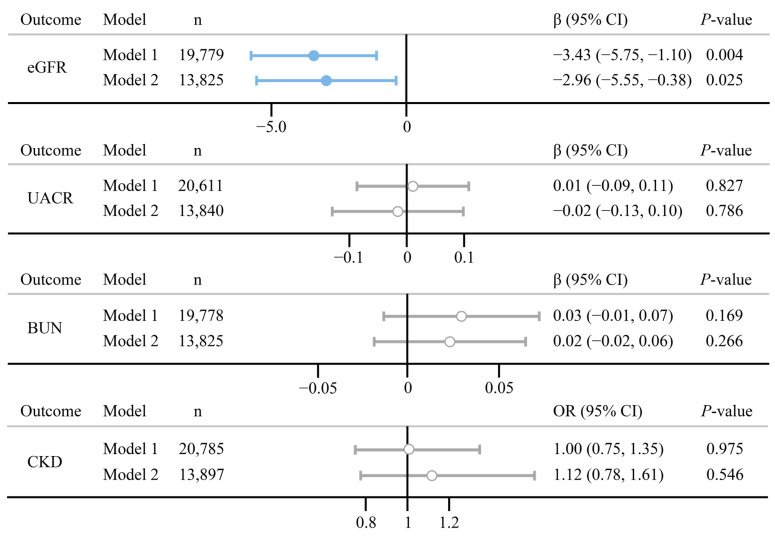
Association between psoriasis and the four renal functions. Model 1 is adjusted for age and gender; Model 2 is adjusted for age, gender, education levels, family PIR, marital status, smoking status, alcohol use, BMI diabetes, and hypertension. n indicates the final number of participants included in the model.

**Figure 3 biomedicines-12-00249-f003:**
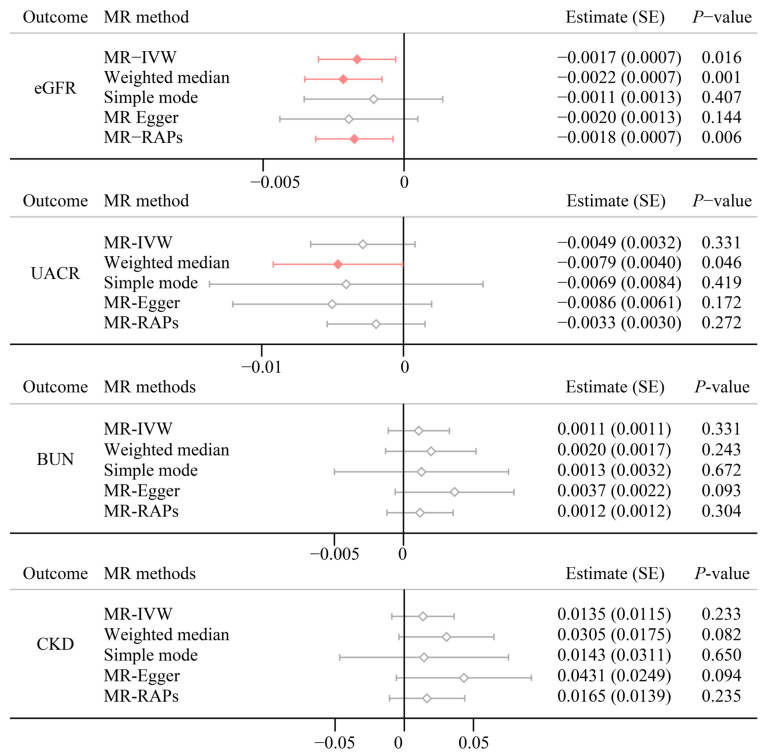
Multi MR method results for psoriasis on renal function. eGFR, estimated glomerular filtration rate based on creatinine levels; UACR, urine albumin to creatinine ratio; BUN, blood urea nitrogen.

**Figure 4 biomedicines-12-00249-f004:**
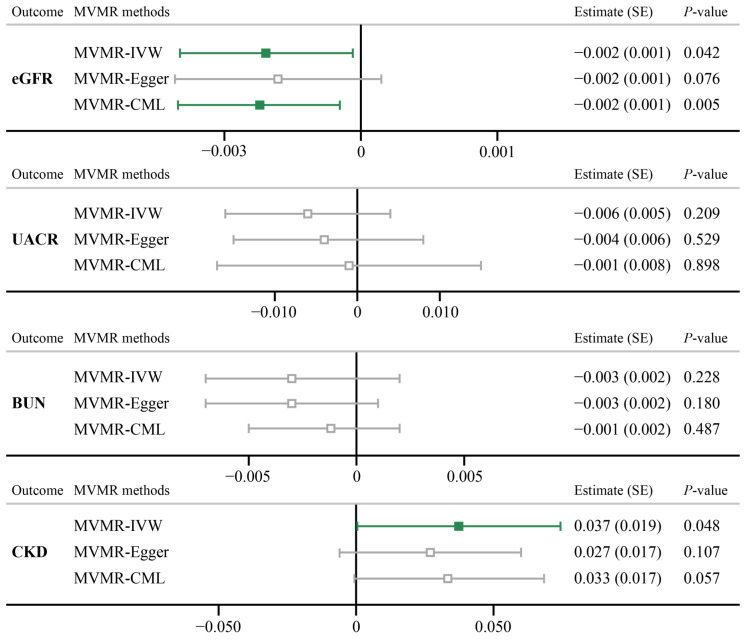
MVMR analyses results for psoriasis on four renal functions. eGFR, estimated glomerular filtration rate based on creatinine levels; UACR, urine albumin to creatinine ratio; BUN, blood urea nitrogen.

**Table 1 biomedicines-12-00249-t001:** Comparison of unweighted population baseline characteristics by psoriasis, NHANES 2003–2006 and 2009–2014 (n = 20,944).

	Overall	Psoriasis		*p*-Value
		No	Yes	
N	20,944	20,395 (97.4)	549 (2.6)	
Age, n (%)				0.223
20~60	16,739 (79.9)	16,312 (80.0)	427 (77.8)	
≥60	4205 (20.1)	4083 (20.0)	122 (22.2)	
Gender, n (%)				0.731
Male	10,738 (51.3)	10,461 (51.3)	277 (50.5)	
Female	10,206 (48.7)	9934 (48.7)	272 (49.5)	
BMI				<0.001
Mean (SD)	28.92 (6.91)	28.89 (6.91)	30.00 (6.99)	
Race/Ethnicity, n (%)				<0.001
White	8735 (41.7)	8410 (41.2)	325 (59.2)	
Black	4761 (22.7)	4687 (23.0)	74 (13.5)	
Mexican American	3425 (16.4)	3379 (16.6)	46 (8.4)	
Other Race	4023 (19.2)	3919 (19.2)	104 (18.9)	
Ratio of family income to poverty, n (%)				0.435
<1.0	4344 (22.5)	4237 (22.5)	107 (21.0)	
≥1.0	14,961 (77.5)	14,558 (77.5)	403 (79.0)	
Education level, n (%)				0.052
Less Than 9th Grade	2066 (9.9)	2029 (10.0)	37 (6.7)	
9–11th/12th grade with no diploma	3129 (15.0)	3054 (15.0)	75 (13.7)	
High School Grad/GED or Equivalent	4810 (23.0)	4686 (23.0)	124 (22.6)	
Some College or AA degree	6281 (30.0)	6107 (30.0)	174 (31.7)	
College Graduate or above	4633 (22.1)	4494 (22.1)	139 (25.3)	
Marital status, n (%)				0.068
Married/Living with partner	12,289 (58.7)	11,958 (58.7)	331 (60.3)	
Widowed/Divorced/Separated	4153 (19.8)	4032 (19.8)	121 (22.0)	
Never married	4490 (21.5)	4393 (21.6)	97 (17.7)	
Smoking status, n (%)				0.641
NO	14,087 (72.9)	13,722 (72.9)	365 (71.9)	
Yes	5248 (27.1)	5105 (27.1)	143 (28.1)	
Alcohol use per day				0.221
median [IQR]	0.10 [0.01, 0.57]	0.10 [0.01, 0.57]	0.07 [0.01, 0.57]	
Hypertension, n (%)				0.001
NO	10,268 (53.6)	10,038 (53.8)	230 (46.2)	
Yes	8875 (46.4)	8607 (46.2)	268 (53.8)	
Diabetes, n (%)				0.035
No	17,845 (85.2)	17,395 (85.3)	450 (82.0)	
Yes	3097 (14.8)	2998 (14.7)	99 (18.0)	
eGFR				0.001
mean (SD)	96.94 (21.25)	97.02 (21.24)	93.96 (21.52)	
UACR				0.147
median [IQR]	6.49 [4.19, 12.13]	6.48 [4.19, 12.07]	6.81 [4.28, 13.94]	
BUN				0.016
mean (SD)	12.68 (5.47)	12.67 (5.45)	13.26 (6.18)	
CKD, n (%)				0.351
No	17,438 (86.1)	16,991 (86.2)	447 (84.7)	
Yes	2806 (13.9)	2725 (13.8)	81 (15.3)	

Note: eGFR, estimated glomerular filtration rate based on creatinine levels; UACR, urine albumin-creatinine ratio; BUN, blood urea nitrogen.

**Table 2 biomedicines-12-00249-t002:** MR-PRESSO results for psoriasis on four renal functions.

Exposure	MR Methods	IVs	Estimate	SE	*p*-Value
eGFR					
	MR-PRESSO (Raw)	31	−0.0016	0.0006	0.023
	MR-PRESSO (Removed outliers)	25	−0.0012	0.0006	0.035
UACR					
	MR PRESSO (Raw)	31	−0.0049	0.0032	0.132
	MR-PRESSO (Removed outliers)	26	−0.002	0.0031	0.520
BUN					
	MR-PRESSO (Raw)	31	0.0011	0.0012	0.370
	MR-PRESSO (Removed outliers)	-	-	-	-
CKD					
	MR-PRESSO (Raw)	31	0.0137	0.0132	0.307
	MR-PRESSO (Removed outliers)	-	-	-	-

Note: The correct-outlier MR-PRESSO was only conducted when *p*-value for MR-PRESSO global test was lower than 0.05. Six outliers (rs1144709, rs12206050, rs1592410, rs62443225, rs77509633, rs847) in eGFR and five outliers (rs118061248, rs13210419, rs636987, rs74787141, rs847) of UACR were excluded in the corrected MR-PRESSO (*p*-value for outlier test <1.000).

## Data Availability

The datasets analyzed for this study are publicly available and accessed as described below. No new data were generated. The data for the observational study were retrieved from the NHANES website: https://www.cdc.gov/nchs/nhanes/index.html (accessed on 7 April 2023). The summary level GWAS data were collected from FinnGen, CKDGen, and UK Biobank, and the corresponding studies can be found in Appendix A.

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
