# Peer review of "Association between Psoriasis and Renal Functions: An Integration Study of Observational Study and Mendelian Randomization"

_biomedicines, 2024, doi:10.3390/biomedicines12010249_

Round 1

Reviewer 1 Report

Comments and Suggestions for Authors

Dear editors:  

 It is a great honor and pleasure for me to be invited as the reviewer for this important work entitled “Association between Psoriasis and Renal Functions: An Integration Study of Observational Study and Mendelian Randomization”. Yuxuan Tan and co-authors comprehensively investigated the recent evidence of relationships and causality between psoriasis and kidney functions, including the estimated glomerular filtration rate (eGFR), blood urea nitrogen (BUN), urine albumin-creatinine ratio (UACR), and chronic kidney disease (CKD). The Mendelian randomization (MR) method of using measured variation in genes of known function is their strength to examine the causal effect of a modifiable exposure on disease in above issues. This study topic is novel and advanced, attributing to corresponding authors’ long-term efforts and contributions in this scientific field. Although the manuscript is well-written, I have a few comments concerning this study:

   Line 63: end-stage kidney disease (ESRD) => ESKD or ESRD?

   Line 114: (eGFR) should be placed behind “estimated glomerular filtration rate”.

The conclusion section is weak that should highlight the bidirectional relationship between psoriasis and CKD parameters: Early detection of CKD in psoriasis patients is an extremely important measure for outcome prevention. Concerning the bidirectional relationship between psoriasis and CKD progression, multidisciplinary shared care plans and complementary therapeutic solutions are necessary for better control of this interdependent pathogenic topic.

The research is interesting that should be published after appropriate revision.

Comments on the Quality of English Language

None.

Author Response

Reviewer #1:

It is a great honor and pleasure for me to be invited as the reviewer for this important work entitled “Association between Psoriasis and Renal Functions: An Integration Study of Observational Study and Mendelian Randomization”. Yuxuan Tan and co-authors comprehensively investigated the recent evidence of relationships and causality between psoriasis and kidney functions, including the estimated glomerular filtration rate (eGFR), blood urea nitrogen (BUN), urine albumin-creatinine ratio (UACR), and chronic kidney disease (CKD). The Mendelian randomization (MR) method of using measured variation in genes of known function is their strength to examine the causal effect of a modifiable exposure on disease in above issues. This study topic is novel and advanced, attributing to corresponding authors’ long-term efforts and contributions in this scientific field. Although the manuscript is well-written, I have a few comments concerning this study:

Reply: We sincerely appreciate your professional review of our manuscript, and we have carefully considered the comments to improve and clarify the manuscript. All comments have been addressed point by point accordingly.

Comments 1: Line 63: end-stage kidney disease (ESRD) => ESKD or ESRD?

The authors' answer: Thank you for pointing out the error. We apologize for the mistake in spelling “end-stage renal disease (ESRD)”. The mistake has been corrected accordingly in the revised manuscript (Page 2, Line 62-63).

Comments 2: Line 114: (eGFR) should be placed behind “estimated glomerular filtration rate”.

The authors' answer: Thank you for your comments. We have move the abbreviation “eGFR” behind the full name “estimated glomerular filtration rate" in the revised manuscript (Page 3, Line 113).

Comments 3: The conclusion section is weak that should highlight the bidirectional relationship between psoriasis and CKD parameters: Early detection of CKD in psoriasis patients is an extremely important measure for outcome prevention. Concerning the bidirectional relationship between psoriasis and CKD progression, multidisciplinary shared care plans and complementary therapeutic solutions are necessary for better control of this interdependent pathogenic topic.

The authors' answer: Thank you for your suggestion. We have revised the conclusion to highlight the causality between psoriasis and CKD parameters (Page 13, Line 377-383).

Comments 4: The research is interesting that should be published after appropriate revision.

The authors' answer: We appreciate your feedback and are pleased to hear that the research is interesting. Thanks again.

Reviewer 2 Report

Comments and Suggestions for Authors

D please, puear Authors,

Well done. Clear presentation of epidemiologic manuscript with well addressed aim, easily readable text and overall scienifically sound presentation. I have just some very small comments:

1) please, separate the Limitations what you mentioned from the Line 361 as separate paragraph;

2) Conclusioons. Please, remove the aim and extra words here, make the conslusions shorten and more precise.

3) go throuth carefully to all the Literature sources - something is lost for the No17 (check also all others for the equal style) and perhaps I can advice to remove or re-place the source 51 from the previous century. You know, it do doesnt fit for this present manuscript more, sorry...

Author Response

Reviewer #2:

Well done. Clear presentation of epidemiologic manuscript with well addressed aim, easily readable text and overall scientifically sound presentation. I have just some very small comments:

Reply: Thank you for your comments and constructive suggestions. We have carefully addressed each of your comments and made necessary revise to the  manuscript.

Comments 1: please, separate the Limitations what you mentioned from the Line 361 as separate paragraph.

The authors' answer: Thank you for your suggestion. We have separated the paragraph related to limitations (Page 12, Line 363).

Comments 2: Conclusions. Please, remove the aim and extra words here, make the conclusions shorten and more precise.

The authors' answer: Thank you for your suggestion. We have removed the aim and extra words from the conclusions, and have shorten them to make them more precise (Page 13, Line 377-383).

Comments 3: go through carefully to all the Literature sources - something is lost for the No17 (check also all others for the equal style) and perhaps I can advise to remove or re-place the source 51 from the previous century. You know, it doesn’t fit for this present manuscript more, sorry...

The authors' answer: Thank you for your suggestion. We have addressed the issue with Reference No. 17 by completing the lost information (Page 14) and have updated Reference No. 51 to more recent evidence to ensure consistency throughout the manuscript (Page 14).

Reviewer 3 Report

Comments and Suggestions for Authors

Dear authors,

Congratulation for your hard work. I have read with great interest your paper and I can say it is a very strong work. I only have one question: why have you chosen data only from 5 NHANES cycles? Why that particular 5 since they are not calendaristical succesive and are quite old. I think you should explain that.

Another minor thing is on line 40. The phrase ends in "clinical". I think you wanted to say "clinical studies" or "clinical trials".

Author Response

Reviewer #3:

Congratulation for your hard work. I have read with great interest your paper and I can say it is a very strong work.

Reply: Thank you very much for your kind words and positive feedback on our manuscript. We are delighted to hear that you found our work to be strong and interesting. We have addressed all your comments and suggestions, and we believe that the revisions have strengthened the manuscript.

Comments 1: I only have one question: why have you chosen data only from 5 NHANES cycles? Why that particular 5 since they are not calendaristical succesive and are quite old. I think you should explain that.

The authors' answer: Thank you for your comments. The choice of data from only five specific NHANES cycles (i.e., 2003-2006 and 2009-2014) was based on the data availability. The selected cycles are not strictly calendaring successive and may seem old, but they provide a substantial sample for epidemiological studies. We acknowledge that including more recent data could provide a more comprehensive understanding of the disease trends and potential confounders. However, the availability of data limited our choice of cycles. We agree that it is important to consider the limitations associated with older data and appreciate your suggestion to clarify this aspect in the manuscript (Page 11, Line 365-368).

Comments 2: Another minor thing is on line 40. The phrase ends in "clinical". I think you wanted to say "clinical studies" or "clinical trials".

The authors' answer: Thank you for your comments. We apologize for the unclear expression in this sentence and we meant to say “clinical cares”. We have made the necessary changes to clarify the expression (Page 1, line 39)